# A Structural Model for Bax∆2-Mediated Activation of Caspase 8-Dependent Apoptosis

**DOI:** 10.3390/ijms21155476

**Published:** 2020-07-31

**Authors:** Bing Xie, Qi Yao, Jialing Xiang, David D.L. Minh

**Affiliations:** 1Department of Chemistry, Illinois Institute of Technology, Chicago, IL 60616, USA; bxie4@hawk.iit.edu; 2Department of Biology, Illinois Institute of Technology, Chicago, IL 60616, USA; qyao2@hawk.iit.edu (Q.Y.); xiang@iit.edu (J.X.)

**Keywords:** apoptosis, Bax, caspase, cell death, computational biology, homology modeling, protein docking, molecular dynamics (MD) simulation

## Abstract

Bax∆2 is a pro-apoptotic anti-tumor protein in the Bax family. While most of the Bax family causes cell death by targeting mitochondria, Bax∆2 forms cytosolic aggregates and activates caspase 8-dependent cell death. We previously showed that the Bax∆2 helix α9 is critical for caspase 8 recruitment. However, the interaction between these two proteins at the structural level is unknown. In this in silico study, we performed molecular dynamics (MD) simulations and protein–protein docking on Bax∆2 variants. The results suggest that the Bax∆2 variants have different stable states. Mutating the Baxα mitochondria-targeting signal [L26P/L27P] appears to introduce a kink into helix α1. Protein–protein docking suggests that helices α9 of both wild-type Bax∆2 and Bax∆2 caspase 8 binding-deficient mutant [L164P] can fit in the same caspase 8 binding site, but the mutant is unable to fit as well as wild-type Bax∆2. Together, these data point to a structural basis for explaining Bax∆2 function in caspase 8-dependent cell death.

## 1. Introduction

Programmed cell death, also called apoptosis, plays a fundamental role in both physiological and pathological processes such as development, tumorigenesis and neurodegeneration [1,2]. Apoptosis occurs through two well-defined apoptotic pathways: extrinsic and intrinsic. The extrinsic pathway is initiated by binding of a cell death ligand to a cell surface receptor, forming a death-inducing signaling complex (DISC). The DISC recruits caspase 8, a cysteine protease, through its death effector domain (DED) [3,4,5]. The intrinsic pathway is triggered by a death insult, such as a chemo-drug, radiation, or intracellular stress. In this pathway, cytosolic Bax monomers oligomerize and form ring-like structures on mitochondria. This causes the release of cytochrome C from the mitochondria and activates the caspase 9 death pathway [6,7,8]. The two pathways eventually merge together through activation of downstream caspases.

Bax is well-known to be a key component in the intrinsic apoptotic pathway and is ubiquitously expressed in almost all human organs [7]. The role of Bax in cancers, autoimmune diseases, and neurodegenerative diseases has been extensively studied [9,10,11]. Bax has a family of isoforms that are mostly generated by alternative splicing [12,13,14]. The parental Bax (Baxα) has 6 exons which encode 9 alpha helices. The evidence suggests that helix α1, mostly encoded by exon 2, contains the mitochondrial targeting signal. In a cell-based assay, a deletion of or a point mutation in helix α1 impairs the mitochondria-targeting ability of Baxα [15,16]. The core part of Baxα is composed of helices α2 to α6, encoded by exon 3 to 5, which contains the “killing” domain and is also required for Baxα oligomerization. The function of Bax helix α9, encoded by the last exon (exon 6), has been controversial but extensive evidence shows that it is responsible for mitochondrial anchoring [17,18].

Bax∆2 is an isoform of the Bax family and a potential target for cancer drugs [19]. Compared to Baxα, Bax∆2 lacks exon 2 and therefore helix α1. Bax∆2 was originally observed in cancer patients with genetic instability [14]. In cancer cells, endogenous Bax∆2 protein is unstable and susceptible to proteasomal degradation [19]. FDA-approved proteasome inhibitor therapeutics, such as bortezomib and carfilzomib, can block degradation of Bax∆2 in cancer cells and promote cell death [20]. Indeed, the pro-death potency of Bax∆2 in cancer cells is significantly higher than that of Baxα [20]. Thus, stabilizing Bax∆2 is a potential strategy for cancer treatment.

In addition to its potential benefits in cancer treatment, Bax∆2 is also compelling because, unlike its relatives in the Bax family, it is not involved in the intrinsic apoptotic pathway, but actually activates the extrinsic pathway [20,21,22]. Like the parental Baxα, Bax∆2 is capable of forming homodimers and a heterodimer with Bcl-2 [14,20]. However, it is unable to attack the mitochondria to activate cell death. Instead, Bax∆2 proteins accumulate in the cytosol and form large protein aggregates, which recruit and activate caspase 8 for cell death [20,21,22]. Helix α9 appears to be essential in this activation process. Deleting helix α9 abolishes the recruitment of caspase 8. Moreover, a point-mutation in helix α9, L164P, significantly impairs the ability of Bax∆2 to recruit caspase 8 for cell death [21].

The contrasting cellular behaviors of Bax variants led us to wonder whether they have distinct structures. Previously, we had predicted protein secondary structure changes with the Netsuvfp server [21]. In these previous calculations, the mutation [L26P/L27P] of Baxα was predicted to significantly change the helicity of the helix α1. Likewise, the mutation [L164P] of Bax∆2 was predicted to disrupt the helicity of helix α9 [21].

In the present contribution, we report modeling at the level of tertiary structure. Although the structure of Baxα has been extensively studied, no structures of Bax∆2 are publicly available. To provide a structural basis for the behavior of Bax∆2, we built models of four Bax variants that were characterized in our previous paper [21]: wild-type Baxα, Bax∆2, Baxα with a mitochondria-targeting mutant [L26P/L27P], and Bax∆2 with a caspase 8 binding mutant [L164P]. The characteristics of all variants are summarized in Figure 1. The models were initially built with RaptorX using a Baxα crystal structure as a template and refined by three independent repetitions of 200 ns of molecular dynamics (MD) simulations. We also performed protein–protein docking to estimate the binding affinities between Bax∆2 and caspase 8 DED. Based on these calculations, we have developed a structural model for the role of Bax∆2 function in caspase 8-dependent cell death.

## 2. Results

### 2.1. Bax Variants Are Predicted to Have Distinct Structures

Due to the helix deletion, we thought Bax∆2 would have a different structure from Baxα variants. However, structures of Bax variants predicted by the RaptorX server were all very similar to one another; this result is probably due to the models being based on the same templates. For example, the predicted structure of Baxα is very similar to its crystal structure (Appendix A). Over the course of 200 ns of MD simulation, distinctions between the variants become more clear (Figure 2, Appendix A). In the simulations, the core parts of Baxα and Baxα[L26P/L27P] remain stable, with a final root mean square deviation (RMSD ) of around 3 Å across all three repetitions. On the other hand, the RMSD of Bax∆2 and its L164P mutant is generally between 3.5 to 6 Å. An exception occurs in one Bax∆2 repetition, in which the structure remains close to the initial configuration.

Representative configurations from the MD simulations provide greater detail about the structural predictions (Figure 2B). Except in helix α1, the tertiary structures of Baxα and Baxα[L26P/l27P] are very similar. Interestingly, the secondary structure elements in Bax∆2 adopt distinct spatial arrangements.

Predicted structural differences between Baxα and Baxα[L26P/L27P] are most evident in helix α1. Over the course of 200 ns of simulation, the RMSD of Baxα helix α1 is mostly within 2 Å of the initial structure. However, in two of three simulations, the RMSD of helix α1 in Baxα[L26P/L27P] dramatically increases within the first 20 ns (Figure 3A) or 125 ns of the simulation (Appendix A) and stays around 3.5 Å. This transition does not occur in the third repetition. Representative structures from the first simulation show that the L26P/L27P mutation causes helix α1 to suddenly kink at position L25 (Figure 3B). Time series of the L25 χ1 dihedral angle show that the increase in RMSD is connected to a change in this angle (Appendix A). In spite of the kink, individual segments of the disrupted helix have comparable stability to wild-type Baxα, with an RMSD mostly under 1.5 Å (Appendix A). Thus, the kink in helix α1 is the likely reason that the L26P/L27P mutant is unable to target mitochondria.

In contrast to the mutant [L26P/L27P], the predicted effects of the mutant [L164P] are less straightforward. The most intuitive explanation of the effects of L164 would be that the mutant disrupts helix α9, interfering with the ability of Bax∆2 to recruit caspase 8. However, the RMSD of helix α9 converges to within 3 Å for all the Baxα and Bax∆2 variants (Figure 4A, Appendix A). The helix does not appear to lose its structural integrity (Figure 4B). In fact, it actually is extended from 18 to 22 residues. Rather than destabilizing helix α9, the mutant appears to affect the structure elsewhere in the protein. In two of three repetitions, a structural perturbation induced by the mutant [L164P] is predicted to be most evident in helix α2 (Figure 5 and Appendix A). A comparable disruption is not observed in a third repetition (Appendix A). For helices α4 to α7, the RMSD of the mutant [L164P] fluctuates within the range observed in other Bax variants (Appendix A). For helix α3, the RMSD of the mutant [L164P] is somewhat larger than with the other variants, while in two of three repetitions, the RMSD of helix α2 converges to a much larger value for the mutant [L164P] (Figure 5A). Indeed, helix α2 appears to be rotated to be nearly perpendicular to its structure in other variants.

The perturbation of helix α2 is predicted to occur through an indirect mechanism. The Bax∆2[L164P] mutant has a smaller side chain, allowing Y98 to move toward helix α9 (Figure 6A,B). This movement is especially evident from its χ2 dihedral angle (Appendix A). The movement of Y98 is correlated with a rotation of helix α4 and movement of F97 towards helix α2 (Figure 6A,C), as also evident in the altered distribution of χ2 of F97 (Appendix A). Because there are hydrophobic contacts among F97, L42 and L46, moving F97 towards helix α2 also causes L42 and L46 to rotate (Figure 6C,D). These rotations stretch helix α2 and further translate helix α3 relative to wild-type Bax∆2.

### 2.2. Helix α9 May Be Involved in Bax∆2-Caspase 8 Docking

Protein–protein docking predicts a direct physical interaction between Bax∆2, Bax∆2[L164P] and the caspase 8 DED. Although Bax∆2 and Bax∆2[L164P] were allowed to dock in any orientation, the lowest-energy docking poses feature an interaction between helix α9 and the DED (Figure 7A).

The calculations are consistent with the experimental observation that Bax∆2[L164P] is unable to mediate caspase 8-dependent apoptosis. The docking score of Bax∆2, −32.982 REU (Rosetta energy units), is lower than the score of Bax∆2[L164P], −29.307 REU, indicating a stronger binding affinity. Hydrogen bonds contribute to the difference in scores. Bax∆2 is predicted to form two hydrogen bonds with the caspase 8 DED: Bax∆2 Q154 with DED E116 (2.191 Å) and Bax∆2 T155 with DED S119 (1.572Å) (Figure 7B). However, no hydrogen bonds were predicted between Bax∆2[L164P] and the caspase 8 DED (Figure 7B). Indeed, all pairwise atomic distances between Bax∆2[L164P] and the caspase 8 DED are larger than 3 Å.

A possible reason that Bax∆2[L164P] does not dock as well as Bax∆2 to the caspase 8 DED is the inability to find the same low-energy pose during the docking calculation. To test whether the binding pose of Bax∆2 bound to the caspase 8 DED was suitable for binding Bax∆2[L164P], we superposed the Bax∆2[L164P] structure with the lowest docking score onto the structure of Bax∆2 in the predicted complex. This superposition leads to steric clashes between three residues, N89, T155, F159, on Bax∆2[L164P] corresponding to E127, S119 and F120 on the caspase 8 DED (Figure 8), indicating that the search algorithm was not the limiting factor in docking Bax∆2[L164P] to the caspase 8 DED.

The relative populations of the representative snapshots also suggests that Bax∆2 is a stronger binder of the caspase 8 DED than its mutant. With Bax∆2, the snapshot with the lowest docking score for the caspase 8 DED is in a highly populated region in the 2D histogram of principal component analysis (Appendix A). On the other hand, the Bax∆2[L164P] snapshot with the lowest docking score for the caspase 8 DED is a minor conformation. Because the binding process must overcome the strain of achieving a minor conformation, the binding affinity of a minor conformation would be weaker even if the docking scores were equivalent.

## 3. Discussion

Using MD simulations and protein–protein docking, we have unveiled details about the likely effect of mutations on the structure of Bax variants and their interactions with caspase 8. The calculations not only explain previous experimental results, but also provide insights that can help design future experiments.

MD simulations support the prevailing hypothesis that helix α1 is essential for mitochondrial targeting [15,16,21]. An alternative hypothesis is that helix α9 is capable of anchoring in the mitochondria without helix α1. The results supporting these hypotheses, however, could depend on the assay systems and cellular context [17,18]. Previously, we showed that the point mutations [L26P/L27P] were predicted to disrupt helix α1 and that they could abolish Baxα translocation to mitochondria [21]. While the mutations are on helix α1, there is a possibility that they can also disrupt helix α9. The present simulations of the [L26P/L27P] mutant predict that helix α1 is indeed perturbed. This prediction could be experimentally validated by hydrogen–deuterium exchange measurements that show greater exchange in portions of helix α5 and the second half of helix α1. The [L26P/L27P] simulations further predict that the structure of helix α9 is not disrupted. Similarly, simulations of the Bax∆2 isoform predict that helix α9 is largely preserved. Thus, disrupting helix α1 is the reason that the mutant and isoform abolish mitochondrial translocation and activation of the mitochondria-dependent intrinsic apoptosis pathway.

Simulations of Bax∆2 also provide structural models that can be used in the computer-aided design of chemical probes and drugs. Presently, no compounds are known to selectively bind to the Bax∆2 isoform. It is possible that ligands which bind Bax∆2 can inhibit its proteasomal degradation and activate apoptosis in cancer cells.

Our modelling also provides a plausible structural mechanism for the interaction between Bax∆2 and caspase 8. Unlike most of the Bax family, Bax∆2 lacks helix α1 and is unable to target the mitochondria [20,21,22]. Instead, it accumulates in the cytosol and forms aggregates. These aggregates serve as a platform recruiting caspase 8 and triggering caspase 8-dependent cell death [21]. Previously, we showed that either performing a mutation on helix α9 or deleting exon 6, which includes helix α9, abolishes the ability of Bax∆2 to bind caspase 8 [21]. However, these previous experiments did not provide detailed structural information about the interaction.

The new calculations suggest that there is a direct physical interaction between helix α9 of Bax∆2 and the caspase 8 DED. In particular, we propose a pair of hydrogen bonds between Bax∆2 Q154 and T155 with caspase 8 DED E116 and S119, respectively. The importance of E116 and S119 on caspase 8 is corroborated by its location on helix α2 of chain B. This helix has been shown to be important in the activation of caspase 8 by dimerization with the DISC [4,23]. Bax∆2 may play a similar role as the DISC in activating caspase 8-dependent apoptosis. The predicted binding mode can be tested by mutations that disrupt the interaction. In particular, we can mutate Q154 and T155 in Bax∆2, which are involved in the proposed hydrogen bonds.

The calculations suggest that our previously designed L164P mutation does not disrupt the structure of Bax∆2 in the most intuitive way. Previously, using protein secondary structure prediction based on NetSurfP 1.1 [24] and GOR4 [25], we predicted that the L164P mutation significantly decreases the helicity of helix α9. This prediction is consistent with rules of thumb regarding the effect of proline in an alpha helix. On the other hand, MD simulations suggest that the mutant actually extends the helix. Moreover, it appears to kink helix α2 (Figure 5) through an indirect mechanism: removing the side chain of L164, allowing rotation of helix α4, and causing F97 to form hydrophobic contacts with L42 and L46 on helix α2 (Figure 6). The MD simulations are more likely than the secondary structure prediction to be correct because they incorporate tertiary structure information.

While the MD simulations suggest that L164P disrupts the structure of Bax∆2 in a counterintuitive way, the subsequent protein–protein docking still supports the importance of helix α9. In Bax∆2, Q154 and T155 are located at the beginning of helix α9 and fit well into the “grove” between two helices of the caspase 8 DED. However, the structural perturbations of Bax∆2[L164P] helix α9, although more subtle than initially anticipated, render it unable to fit in the same way as wild-type Bax∆2.

Considering both our calculations and previous experimental results, we propose that both helix α9 and an aggregate of sufficient size are necessary for the activation of caspase 8-mediated apoptosis by Bax variants [21]. We have previously demonstrated that a Baxα variant without a BH3 domain could form fine uniform aggregates (most likely oligomers), but were unable to recruit caspase 8 even with an intact helix α9 [21]. However, a Bax∆2 mutant without helix α9 can form larger protein aggregates but cannot recruit caspase 8 and induce cell death [21]. Consistently, such large “platform”-mediated activation of caspase 8 has been also observed in other cellular contexts, such as activation of caspase 8 by autophagosome, a large membrane-bound structure, and DISC, a large membraneless protein complex [3,4,5,26].

At this point, the anatomic properties of the Bax∆2 aggregates necessary to recruit caspase 8 remain unclear. Bax∆2 aggregates observed under a microscope after immunostaining vary in both size and shape. We are unable to determine the position of helix α9 in these aggregates. Based on the available crystal structures and our computational modeling, it appears that helix α9 is located on the surface of the protein in monomers of both Baxα and Bax∆2. For Baxα, upon a death signal stimulation, the monomers change conformation and organize into oligomers that anchor into the mitochondrial outer membrane via a barrel channel structure [27]. In this process, helix α1 appears to serve as a mitochondrial targeting signal and leads Bax monomers to mitochondria while helix α9 serves as an “anchor” to facilitate Bax oligomerization and formation of a “pore” on the outer membrane of mitochondria. Previous models of the Bax pore position α2 helices in the inner part of the pore and helices α7 and α8 at the periphery of the pore [28]. Therefore, helix α9 appears to not be a part of the pore structure itself, but essential for Bax pore formation [18,28]. Deletion or mutation of helix α9 could abolish the ability of Bax to permeabilize membranes [27]. However, whether helix α9 is well-organized or randomly presented on the surface of Bax∆2 aggregates remains to be explored. Nevertheless, the surface location of helix α9 seems critical for its accessibility in “reaching out” for recruitment of caspase 8. In the future, properties of these aggregates may be studied by a combination of experimental and computational biophysical methods.

## 4. Materials and Methods

### 4.1. Protein Sequences

The open reading frames from amino acid sequences of Baxα (AAA03619) and Bax∆2 (AFU81108) were obtained from the GenBank database [29]. The amino acid sequence of the Baxα mitochondria-targeting mutant [L26P/L27P] was generated by substituting the leucine with proline at positions 26 and 27. The amino acid sequence of Bax∆2-caspase 8 mutant [L164P] was generated by substituting the leucine with proline at position 164.

### 4.2. Homology Modeling

Initial structural models for the four Bax variants were generated using RaptorX [30], a protein secondary and tertiary structure prediction server. RaptorX predicts the secondary structures, solvent accessibility, and disordered regions for a protein. It then uses this information to search for templates using the Modeller [31] or Rosetta [32] servers and then constructs a 3D model. To construct the models, protein sequences of the four Bax variants were uploaded to RaptorX web server in FASTA format. Server options of no “pred ligand binding” and “pred go term” were selected. For comparison with the known Baxα crystal structure (PDB ID: 1F16), the model of Baxα was also built.

### 4.3. Molecular Dynamics Simulation

For each variant, MD simulations were performed using OpenMM 7.0 [33]. Protein structures were parameterized with the AMBER ff14SB force field [34]. Each system was solvated in a box of TIP3P water molecules with at least 10 Å of padding on each side. Na^+^ and Cl^−^ ions were added to the system for neutralizing and modeling the 0.15 M physiological concentration. Each system was energy-minimized for 500 steps, and each MD simulation was performed in the NPT (constant number of particles, pressure, and temperature) ensemble and was propagated for 200 ns. The pressure was controlled by the Monte Carlo barostat at 1 atmosphere of pressure with moves every 25 time steps. Particle mesh Ewald (PME) was used to treat long-range nonbonded interactions. A Langevin dynamics integrator with a 2 fs time step was used, with a bath temperature of 300 K and constraints on bonds involving hydrogen. Energies and frames were recorded every 1 ps. The simulations were repeated three times for each variant.

### 4.4. Analysis of MD Simulation

Prody 1.8.2 [35] and Visual Molecular Dynamics (VMD) 1.9.0 [36] version were used to analyze MD trajectories. Trajectories were combined through catdcd 4.0 [36]. All snapshots were aligned to the initial structure by minimizing the root mean square deviation with backbone atoms in the protein core: residues 54 to 171 of Baxα or residues 37 to 154 of Bax∆2. The residue numbers corresponding to helix α1 to helix α9 are listed in Table 1.

Smoothing of time series was performed using the Savitzky–Golay filter implemented in the python package scipy 1.2.1 [37]. The filter was run with a window length of 10,001 (out of 100,000) for each individual simulation, and the polynomial order was 3. Other options were deriv = 0, delta = 1.0, axis = −1, mode = ‘interp’, and cval = 0.0.

Principal component analysis (PCA), a dimensionality reduction technique, was performed using scikit-learn decomposition 0.15.2 [38]. First, the Baxα, Baxα[L26P/L27P], Bax∆2 and Bax∆2[L164P] trajectories were aligned to the initial structure of Baxα based on the heavy atoms of the backbone in the core. For all four variants, coordinates from backbone heavy atoms in the core were used in the PCA model. 

Representative structures were selected by constructing a histogram of the first two principal components of each molecular dynamics trajectory. A structure was selected from near the mode of the 2D histogram.

### 4.5. Protein–Protein Docking

K-means clustering with 100 clusters was performed using scikit-learn 0.15.2 [38]. Distances between each pair of snapshots were based on the Euclidean distance between the first two principal components. Cluster centers were picked as representative structures. Representative structures of Bax variants were docked to the caspase 8 death effector domain (DED) (PDB ID:5H33) using Rosetta 2019.22 [39]. The caspase 8 DED was marked as chain A and the Bax variant was marked as chain B. Rosetta parameters used for global protein–protein docking are as listed below: -nstruct 100, -partners A_B, -dock_pert 3 8,-spin, -randomize1, -randomize 2. We picked the lowest-energy pose of each complex for structural analysis.

## Figures and Tables

**Figure 1 ijms-21-05476-f001:**
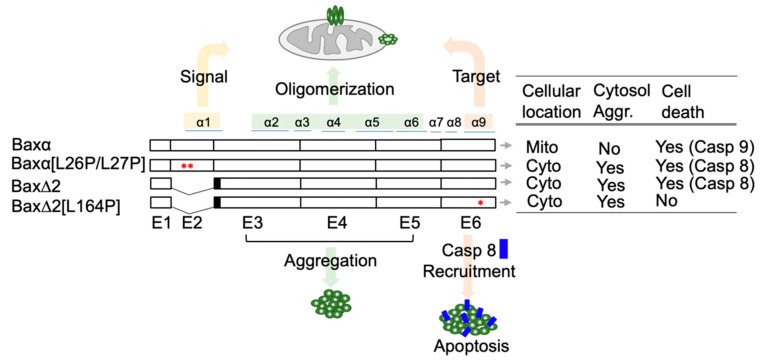
An overview of characteristics of Bax variants. Cellular behaviors of Bax variants. Bax exons are labeled as E1 to E6. The corresponding helices are labeled on the top. The solid black boxes on Bax∆2 and Bax∆2[L164P] indicate the region coding for frameshifted peptide sequences. * indicates the position of mutations, which are adjacent to one another in Baxα[L26P/L27PP]. Mito, mitochondria; Cyto, cytosol; Aggr, aggregation; Casp, caspase.

**Figure 2 ijms-21-05476-f002:**
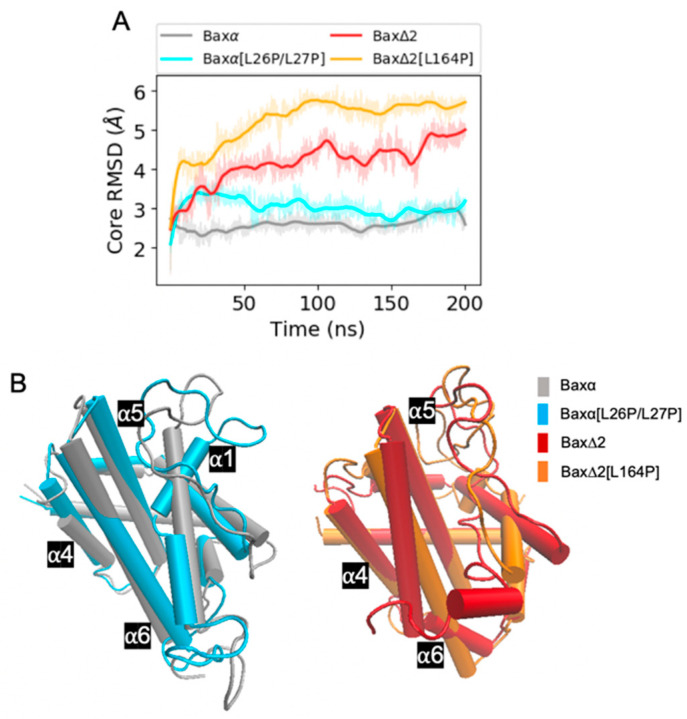
An overview of Bax variant simulations. (**A**) RMSDs of core backbone atoms relative to the initial structure for a period of 200 ns of molecular dynamics (MD) simulations, and (**B**) representative structures selected as described in Section 4.4. Coloring is by variant: Baxα (grey), Baxα[L26P/L27P] (blue), Bax∆2 (red), and Bax∆2 [L164P] (orange). Shaded lines show RMSDs for all recorded points and solid lines are smoothed as described in Section 4.4. Similar plots for two additional repetitions of the simulations are shown in the upper left panels of Appendix A.

**Figure 3 ijms-21-05476-f003:**
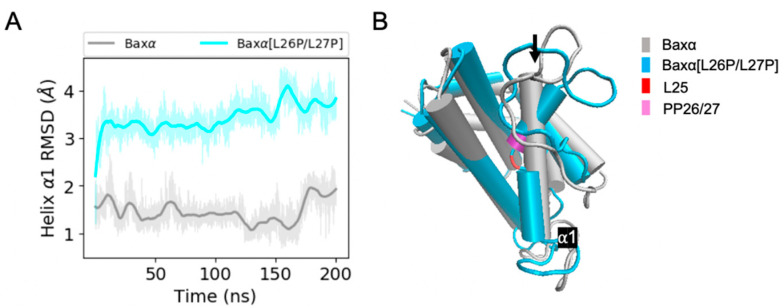
RMSD analysis of helix α1. (**A**) RMSDs of Baxα and Baxα[L26P/L27P] helix α1 backbone atoms in the core region relative to the initial structure for a period of 200 ns of MD simulations. (**B**) Representative structures of Baxα (grey) and Baxα[L26P/L27P] (blue). The black arrow points at helix α1. Shaded lines show RMSDs for all recorded points and solid lines are smoothed as described in Section 4.4. Similar plots for two additional repetitions of the simulations are shown in the upper right panels of Appendix A.

**Figure 4 ijms-21-05476-f004:**
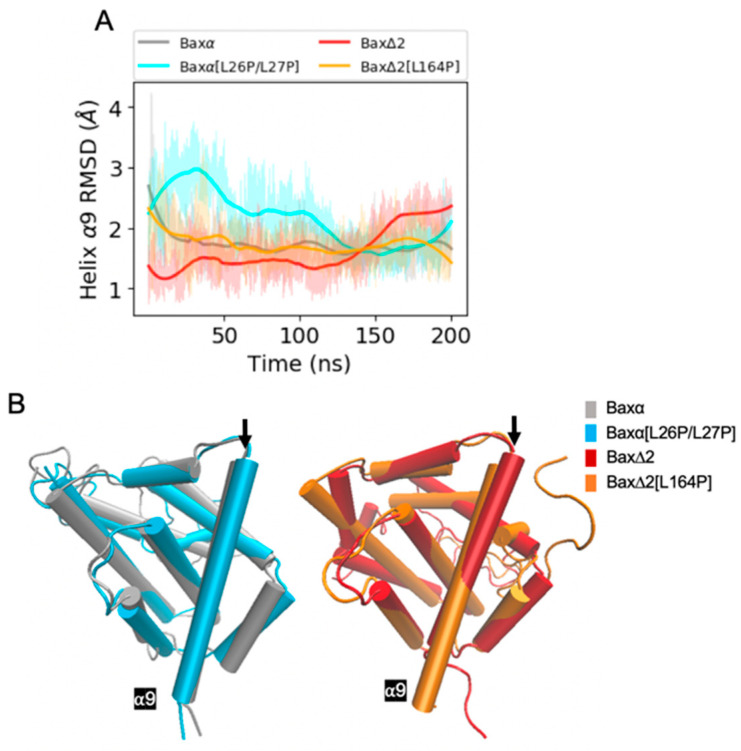
RMSD analysis of helix α9. (**A**) RMSDs of Bax variant helix α9 backbone atoms relative to the initial structure for a period of 200 ns of MD simulations. Shaded lines show RMSDs for all recorded points and solid lines are smoothed as described in Section 4.4. Similar plots for two additional repetitions of the simulations are shown in the bottom right panels of Appendix A. (**B**) Representative structures of Bax variants. The black arrow points at helix α9.

**Figure 5 ijms-21-05476-f005:**
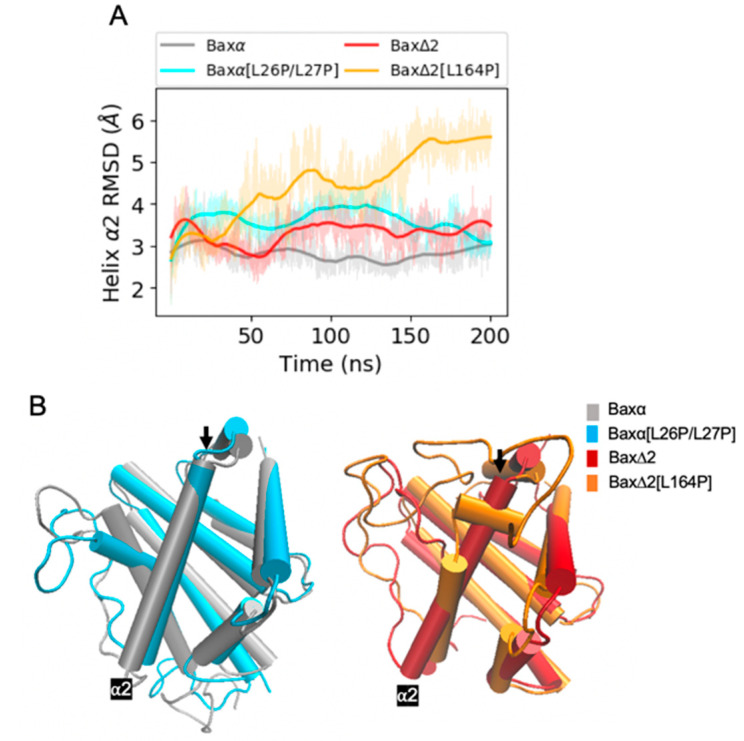
RMSD analysis of helix α2. (**A**) RMSDs of Bax variant helix α2 backbone atoms relative to the initial structure for a period of 200 ns of MD simulations. Shaded lines show RMSDs for all recorded points and solid lines are smoothed as described in Section 4.4. Similar plots for two additional repetitions of the simulations are shown in the bottom left panels of Appendix A. (**B**) Representative structures of Bax variants. The black arrow points at helix α2.

**Figure 6 ijms-21-05476-f006:**
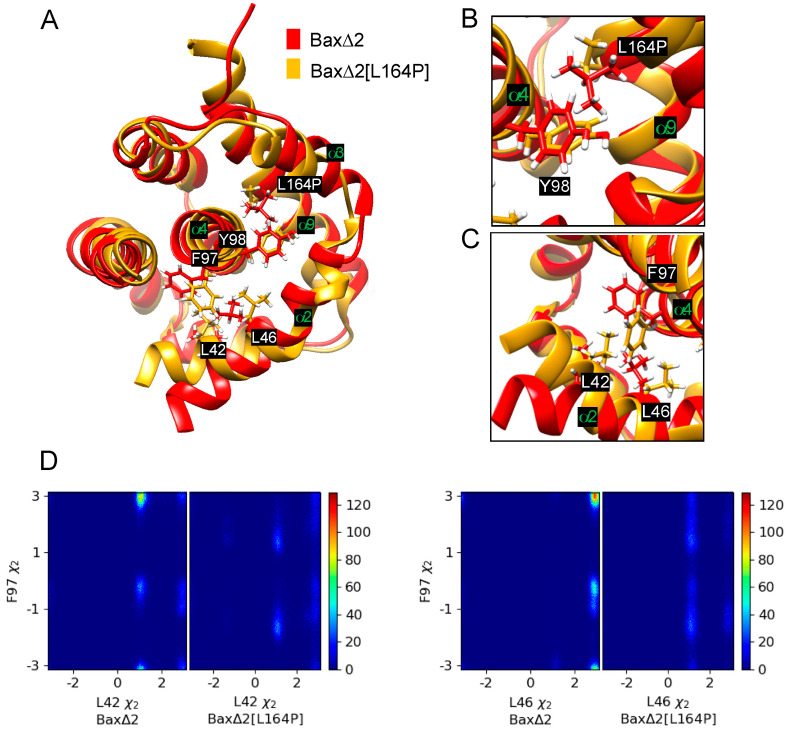
Analysis of dihedral angles. (**A**) Representative structures of Bax∆2 and Bax∆2[L164P]. Several helices and key amino acids of Bax∆2 are labeled. (**B**) Closeup view of the L164P and Y98 region. (**C**) Closeup view of the F97, L42, and L46 region. (**D**) Two-dimensional histograms of dihedral angle pairs based on 200 ns of MD simulation: L42χ2 and F97χ2 (**left**), L46 χ2 and F97 χ2 (**right**).

**Figure 7 ijms-21-05476-f007:**
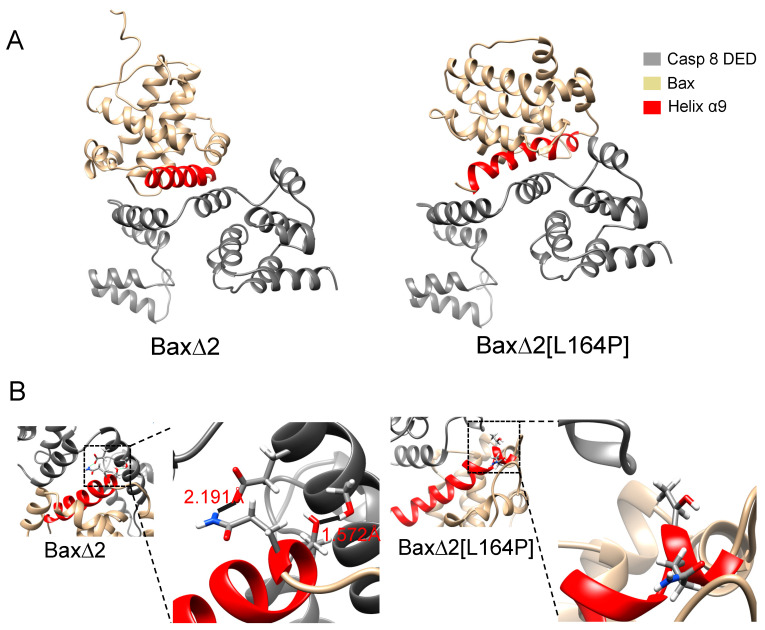
Protein–protein docking poses. (**A**) The lowest-energy binding pose of Bax∆2 with caspase 8 (**left**). The lowest-energy binding pose of Bax∆2 [L164P] with caspase 8 (**right**). (**B**) Details of Bax∆2 binding (**left**) and Bax∆2[L164P] binding (**right**). In all panels, Bax∆ 2 is colored gold, helix α9 red, and the caspase death effector domain (DED) grey.

**Figure 8 ijms-21-05476-f008:**
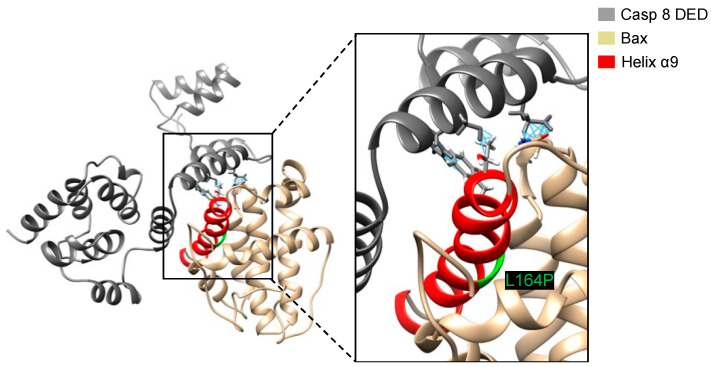
The interaction between Bax∆2 [L164P] and caspase 8 DED after superposing Bax∆2[L164P] to the lowest docking pose of Bax∆2. A closeup view of the binding site is shown on the right. The location of P164 is marked green and close contacts are shown with blue lines.

**Table 1 ijms-21-05476-t001:** Amino acid indexes corresponding to helix indexes.

2nd Structure	Baxα Baxα[L26P/L27P]	Bax∆2 Bax∆2[L164P]
Loop	1–15	1–36
Helix α1	16–35	-
Helix α2	54–72	37–55
Helix α3	74–82	57–65
Helix α4	88–100	71–83
Helix α5	108–127	91–110
Helix α6	130–146	113–129
Helix α7	148–154	131–137
Helix α8	158–164	141–147
Helix α9	171–188	154–171

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
