# Peer review of "A Structural Model for Bax∆2-Mediated Activation of Caspase 8-Dependent Apoptosis"

_ijms, 2020, doi:10.3390/ijms21155476_

Round 1

Reviewer 1 Report

In this manuscript, the authors built the BaxΔ2 protein using homology modeling and then performed molecular dynamics simulations and protein-protein docking on BaxΔ2 variants. Several structural insights were observed in a course of 200ns simulation trajectory, which may shed light on the differences of the BaxΔ2 variants upon caspase 8 recruitment. The manuscript falls into the scope of International Journal of Molecular Sciences. However, several questions need to be answered before considering for publication.

  1. In the manuscript, only 200 ns MD simulation was carried out. That's really very short simulation. It may not provide comperhensive structural insights. For example, the lack of conformational difference for alpha-9 helix in BaxΔ2[L164P] may be due to the shortness of the trajectory. Besides, all the 4 proteins studied in this manuscript (Baxα, BaxΔ2, Baxα[L26P/L27P], BaxΔ2[L164P]), have less than 200 amino acids. For such a small protein system, a V100 GPU on Amber can provide more 200 ns data in one day. Microsecond-length simulation should be very doable and informative for this work.
  2. For the 200ns simulation, did you repeat the simulation several times? Is the result reproducible?
  3. Line 65, although homology modeling were given in section 4.2, the author should make it clear in line 65 on what approach is used to built the protein structures. An example plot to highlight the sequences identity of the aligned structures should be added to help understand the differences.
  4. Line 98, it's interesting to find the kink conformation. When did it show up in the simulation? 
  5. The position of 26 and 27 should be displayed on Figure 2B.
  6. Which amino acid position show the kink? 
  7. Can you make a dihedral plot at the kink position to show the torsion angle changes along the simulation trajectory?
  8. What's the driving force to cause the kink in the simulation?
  9. Can you validate the Kink and the later perpendicular conformations with experiments, such as hydrogen-deuterium exchange?
  10. Line 235, the GeneBank ID for BaxΔ2 protein sequence is wrong. It should be AFU81108.
  11. Line 105, change "most straightforward" to "most intuitive".
  12. So many "on the other hand" in the manuscript. Suggest to choose alternative phrases.

Author Response

> In this manuscript, the authors built the BaxΔ2 protein using homology modeling and then performed molecular dynamics simulations and protein-protein docking on BaxΔ2 variants. Several structural insights were observed in a course of 200ns simulation trajectory, which may shed light on the differences of the BaxΔ2 variants upon caspase 8 recruitment. The manuscript falls into the scope of International Journal of Molecular Sciences. However, several questions need to be answered before considering for publication.

> In the manuscript, only 200 ns MD simulation was carried out. That's really very short simulation. It may not provide comprehensive structural insights. For example, the lack of conformational difference for alpha-9 helix in BaxΔ2[L164P] may be due to the shortness of the trajectory. Besides, all the 4 proteins studied in this manuscript (Baxα, BaxΔ2, Baxα[L26P/L27P], BaxΔ2[L164P]), have less than 200 amino acids. For such a small protein system, a V100 GPU on Amber can provide more 200 ns data in one day. Microsecond-length simulation should be very doable and informative for this work. For the 200ns simulation, did you repeat the simulation several times? Is the result reproducible?

We agree that based on current technology, 200 ns is not a long simulation. For this reason, our analysis was based on three repetitions, to total 600 ns of data. Our analysis suggests that this amount of simulation is sufficient to draw meaningful conclusions. Major structural changes are observed within this time frame and are reproduced across multiple repetitions. To better show this reproducibility, we have added plots related to the second and third repetitions of the simulations as supplementary figures.

> Line 65, although homology modeling were given in section 4.2, the author should make it clear in line 65 on what approach is used to built the protein structures. 

This paragraph refers to a previous paper. To clarify, we have added the phrase “In these previous calculations...”. (Lines 63,68)

> An example plot to highlight the sequences identity of the aligned structures should be added to help understand the differences.

Figure 1 provides an overview of the differences between the Bax variants. The sequences are completely identical except for the noted mutations and deletions.

> Line 98, it's interesting to find the kink conformation. When did it show up in the simulation? The position of 26 and 27 should be displayed on Figure 2B. Which amino acid position show the kink? Can you make a dihedral plot at the kink position to show the torsion angle changes along the simulation trajectory? What's the driving force to cause the kink in the simulation?

Thanks to the reviewer’s comment, we have performed a more thorough analysis of the kink conformation. We have revised Figure 3B to show the positions of L25, the location of the kink, and PP26/27, the location of the mutations. We have created a new supplementary figure, S4, to show the dihedral angle of L25 as a function of time. These figures are now discussed in the manuscript. (Lines 113-135)

> Can you validate the Kink and the later perpendicular conformations with experiments, such as hydrogen-deuterium exchange?

We thank the reviewer for the suggestion. Yes, in principle the kink could be validated by observing greater hydrogen-deuterium exchange at residues between helix α1 and α5. A statement describing this proposed experiment has been added to the discussion section. (Lines 268-272)

> Line 235, the GeneBank ID for BaxΔ2 protein sequence is wrong. It should be AFU81108.

We now provide the GenBank numbers for both the Baxα and Bax∆2 amino acid sequences. (Line 368)

> Line 105, change "most straightforward" to "most intuitive".

We have made the requested change (Line 143).

> So many "on the other hand" in the manuscript. Suggest to choose alternative phrases.

Thank you for pointing this out. In various places, we have changed “on the other hand” to “as expected”, “however”, or “surprisingly”.

Reviewer 2 Report

The authors applied computational modeling to understand structural changes associated to alpha1 disruption (Bax alpha L26P/L27P), alpha1 deletion (Bax Delta2), alpha9 disruption (Bax Delta2 L164P), compared to parental Bax alpha.
They performed 3 200-ns MD simulations in the NPT ensemble for each system.
These changes are used to understand Bax Delta2 recruitment of Casp 8.

The simulations performed by authors show the settling of initial models generated by the RaptorX web server, the latter acting on the template Bax alpha known NMR 1F16 PDB structure.
As expected (lines 61-62), the Leu to Pro mutations disrupt the alpha helices where mutations occur. The MD simulations settle the effects of this disruption.
The simulation of Bax Delta2 and its L164P variant allows the selection of configurations to dock into a rigid model of Casp 8 death effector domain (DED).
The structural changes affecting Bax Delta2 L164P explain the weaker recruitment of Casp 8 by this variant, though only monomeric partners
are here addressed.
The rearrangment of side chains as long-range effect of alpha9 disruption, described as counterintuitive (line 210), is analyzed in detail.

Despite the limitations of the computational approach in terms of statistical accuracy (a few short MD simulations) and sample size (oligomers are not accounted for), this study is useful to understand the modulation of induced apoptosys, as stated by authors at lines 171-172. Therefore, I recommend the
publication in IJMS in a revised form.

Several points that can be clarified are described below.

Fig.1 - I guess some labels are hidden below shadow boxes at the top.

Fig.2 - The right panel can not be understood: this kind of figures is not clear when more than two configurations are displayed. The same holds for Figs.4-5 (right panels, but also in left panels there are too many curves).
There should be at least a little mention of the criteria (see Methods) to choose representative configurations (line 88). Since there are 3 200-ns long simulations, the reader does not understand whether averaging over 3 trajectories is performed in the left panels or not. Line 259 is not clear, please explain how the 3 trajectories (lines 255-256) are combined and in what extent the 3 trajectories differ. Do initial configurations change? If not, the 3 trajectories are likely very similar and statistics is not improved.

In SI Fig.S2 is repeated.
Curves in Fig.S2 must be separated to be understood.

Line 95 - RMSD is repeated. Check syntax in some sentences.

Lines 97-98 - A kink is formed by construction with the replacement of LL with PP.
There may be stabilization of one or both helices fragmented by the kink, as it is the case of alpha9 in Bax Delta2 variant.

Lines 236-239 - The replacement of Leu with Pro is not trivial, as a distortion of Phi angle is constructed. Some more details about the atomic substitution is expected in Methods. Do MD simulations start with extended bonds, keeping Phi unchanged?
The construction described at lines 241-245 is not reproducible.

Table 1, column 2 - 168 is 158.

Line 216 - There is no evidence in the manuscript of the requirement of "an aggregate of sufficient size" for Casp 8 activation.

Line 230 - A barrel channel consisting of helix alpha9 is difficult to conceive. Indeed, ref. 26 does not mention this type of channel structure. However, the role of alpha9 in the mechanism proposed in ref. 26 can be better discussed, even though the mitochondrial membrane modification is out of the scope of the manuscript.

Round 2

Reviewer 1 Report

All the questions are well answered by the authors and revised in the manuscript accordingly. No further review is required.

The manuscript can be accepted for publication in present form.